# Controlling forward and backward rotary molecular motion on demand

L. Pfeifer [1,5], S. Crespi [1,6], P. van der Meulen[1], J. Kemmink [1], R. M. Scheek[1], M. F. Hilbers [2], W. J. Buma [2,3] & B. L. Feringa [1,4✉]

Synthetic molecular machines hold tremendous potential to revolutionize chemical and materials sciences. Their autonomous motion controlled by external stimuli allows to develop smart materials whose properties can be adapted on command. For the realisation of more complex molecular machines, it is crucial to design building blocks whose properties can be controlled by multiple orthogonal stimuli. A major challenge is to reversibly switch from forward to backward and again forward light-driven rotary motion using external stimuli. Here we report a push-pull substituted photo-responsive overcrowded alkene whose function can be toggled between that of a unidirectional 2$^{nd}$ generation rotary motor and a molecular switch depending on its protonation and the polarity of its environment. With its simplicity in design, easy preparation, outstanding stability and orthogonal control of distinct forward and backward motions, we believe that the present concept paves the way for creating more advanced molecular machines.

[1] Stratingh Institute for Chemistry, University of Groningen, Nijenborgh 4, 9747 AG Groningen, The Netherlands. [2] Van't Hoff Institute for Molecular Sciences, University of Amsterdam, Science Park 904, 1098 XH Amsterdam, The Netherlands. [3] Institute for Molecules and Materials, FELIX Laboratory, Radboud University, Toernooiveld 7c, 6525 ED Nijmegen, The Netherlands. [4] Zernike Institute for Advanced Materials, University of Groningen, Nijenborgh 4, 9747 AG Groningen, The Netherlands. [5] Present address: Laboratory of Photonics and Interfaces, Department of Chemistry and Chemical Engineering, École Polytechnique Fédérale de Lausanne, Station 6, Lausanne CH-1015, Switzerland. [6] Present address: Department of Chemistry, Ångström Laboratory, Uppsala University, Box 523, 751 20 Uppsala, Sweden. ✉email: b.l.feringa@rug.nl

Over the last few decades, the field of synthetic molecular machines has attracted steeply increasing attention due to the major impact these machines can have on the design of artificial, dynamic systems controllable by external stimuli and potential applications in areas ranging from biomedical to smart materials[1–12]. Moving from molecules to dynamic molecular systems ultimately offers tremendous opportunities to control, explore and use motion at the nanoscale and beyond. On a functional level, one can draw a distinction between the two subfields of molecular switches and motors, with the former possessing two distinguishable states interconvertible via a back-and-forth type motion[13]. Any work performed during one step is in this case reversed by the subsequent one moving the system back into its original state. Prominent examples of such molecular switches include azobenzenes[14–17], diarylethenes[18], stilbenes[19,20], hydrazones[21] and spiropyrans[22,23] as well as chiroptical switches[24,25]. Taking inspiration from Nature, the introduction of (pseudo-)asymmetry into the backbone of these molecular entities[26] or, alternatively, the use of specific sequences of chemical transformations[27] allowed for the preparation of motors operating in a unidirectional manner. Molecular motors can therefore be used to, among others, induce directional rotary motion and progressively move systems away from equilibrium opening up a plethora of new applications with regard to more complex functional systems[28]. Starting from the controlled rotation around single bonds via subsequent chemical transformations[27,29], one of the most prominent concepts for construction of molecular motors today makes use of the light-driven isomerization of overcrowded alkenes, where unidirectional rotation around the central $C=C$ double bond is achieved via sequential excited state $E/Z$-isomerization and thermal helix inversion (THI) steps[30,31]. Related molecular designs explored by Lehn[32] and Dube[33–35] include imines and hemi-thioindigos. Artificial rotary molecular motors have been used for the preparation of functionalized soft materials[36–39], multitasking catalysts[40], liquid crystals[41], photoresponsive surfaces[42–44] as well as dynamic solid materials[45–47] demonstrating their unique potential for the design of smart materials. Potential applications that have been demonstrated in seminal studies include the creation of artificial muscles[38], molecular motor-controlled movement of microscale objects on functionalized surfaces[42], control over stem cell differentiation[44] as well as photo-activated opening of cell membranes[48]. Recently, we also demonstrated the operation of a 2nd generation rotary motor using low-intensity near-infrared (NIR) light via two-photon absorption[49].

Controlling molecular motion by light offers the advantage of outstanding spatiotemporal control over the use of chemical reactants. However, in order to harness the full potential of molecular motors, it is crucial to be able to exert reversible orthogonal control over their behaviour by different external stimuli. In the past, several studies have demonstrated in situ control over certain motor properties. For example, several successful attempts have been described for modifying the rotational speed of overcrowded alkene-based rotary molecular motors via addition of modulators[50–52]. In some instances, these strategies have simultaneously led to changes in the UV-vis absorption characteristics of these systems[50,52]. Another example where motor function was controlled in situ showed a more efficient photoswitching process following protonation of a NMe₂ group[53]. Our group has also recently described a light-gated molecular motor functionalized with a dithienylethene switch (Fig. 1a)[54]. Another elegant example describes the induction of unidirectional motion induced by supramolecular interaction allowing the in situ transformation of an achiral stiff-stilbene switch into a unidirectional rotary motor (Fig. 1b)[55,56]. Using hemithioindigo-based molecular photoswitches instead of motors Dube demonstrated control over both the reaction coordinate of the photoisomerization[57] as well as the UV-vis absorption properties and speed of thermal back-reaction by changing solvent polarity[58]. These seminal results are an important step to more complex mechanical function and clearly demonstrate the interest in controlling molecular motion via orthogonal stimuli to achieve advanced levels of control over functional systems which will ultimately be assembled from these building blocks.

Here, we demonstrate an approach for controlling molecular motion by external stimuli based on a push–pull type overcrowded alkene. X-ray diffraction, NMR, UV-vis absorption as well as extensive kinetic and Density Functional Theory (DFT) studies show that the mode of operation of this system can be toggled between that of a unidirectional rotary molecular motor and a molecular switch depending on the polarity of its environment and its protonation state (Fig. 1c, d). In fact, we can reversibly control forward (rotary motor function) and backward (molecular switch function) light-driven motion using an orthogonally controlled actuator moiety. With molecular motors, contrary to switches, being able to carry out work upon progressive operation, this could in future be used to apply work only in those parts of a complex system where the conditions for motor function are met allowing more complex machine operations. In cells this could for example be used for highly targeted drug delivery. Furthermore, upon integration into porous materials like MOFs or COFs one could alter their properties by filling the pores with solvents of different polarity.

## Results and discussion

**Molecular design and synthesis**. Motor **1** was designed following the general principle of a push-pull alkene with a strongly electron donating amino group attached to the upper half and two electron withdrawing cyano groups attached to the lower one. Donating and withdrawing groups are in conjugation via the central double bond (rotary axle) in overcrowded alkene **1**. Two five-membered rings were chosen to flank this double bond as this kind of 2nd generation Feringa type motor generally possesses THI half-lives in the range of minutes at room temperature making them convenient for studying this isomerization processes. Figure 2 shows the synthesis of the stable isomer of motor **1** (**1ₛ**) starting from the Barton-Kellogg olefination showing the sequence of steps that we followed in order to successfully introduce the amino as well as cyano groups.

For the Barton-Kellogg olefination, ketone **2**, bearing an acetyl protected phenolic alcohol group, was first converted into its thioketone analogue before it was reacted with 3,6-dibromo-9-diazo-9H-fluorene (**3**). Subsequent sulphur abstraction was achieved using hexaethyl phosphorous triamide (HEPT) to give **4ₛ** with a 42% overall yield, the more commonly used HMPT was found to cause deprotection of the alcohol. It was necessary to perform this olefination before introducing any of the final electron donating or withdrawing groups as their presence in the respective precursors was found to be detrimental to the reaction's outcome. In the next step, the CN groups were introduced by performing a Pd-catalysed cyanation reaching a yield for **5ₛ** of 74%. At this point in the synthesis, the acetyl protecting group was removed and the free phenol subsequently converted into a triflate to give **6ₛ** in an overall yield of 62%. Finally, the pyrrolidino substituent was introduced using a Pd-catalysed Buchwald-Hartwig amination giving **1ₛ** as a dark purple solid in 63% yield. For this final step, it was found to be crucial to work at room temperature and keep the reaction time as short as possible due to the sensitivity of **6ₛ** to strong bases leading to degradation of the motor.

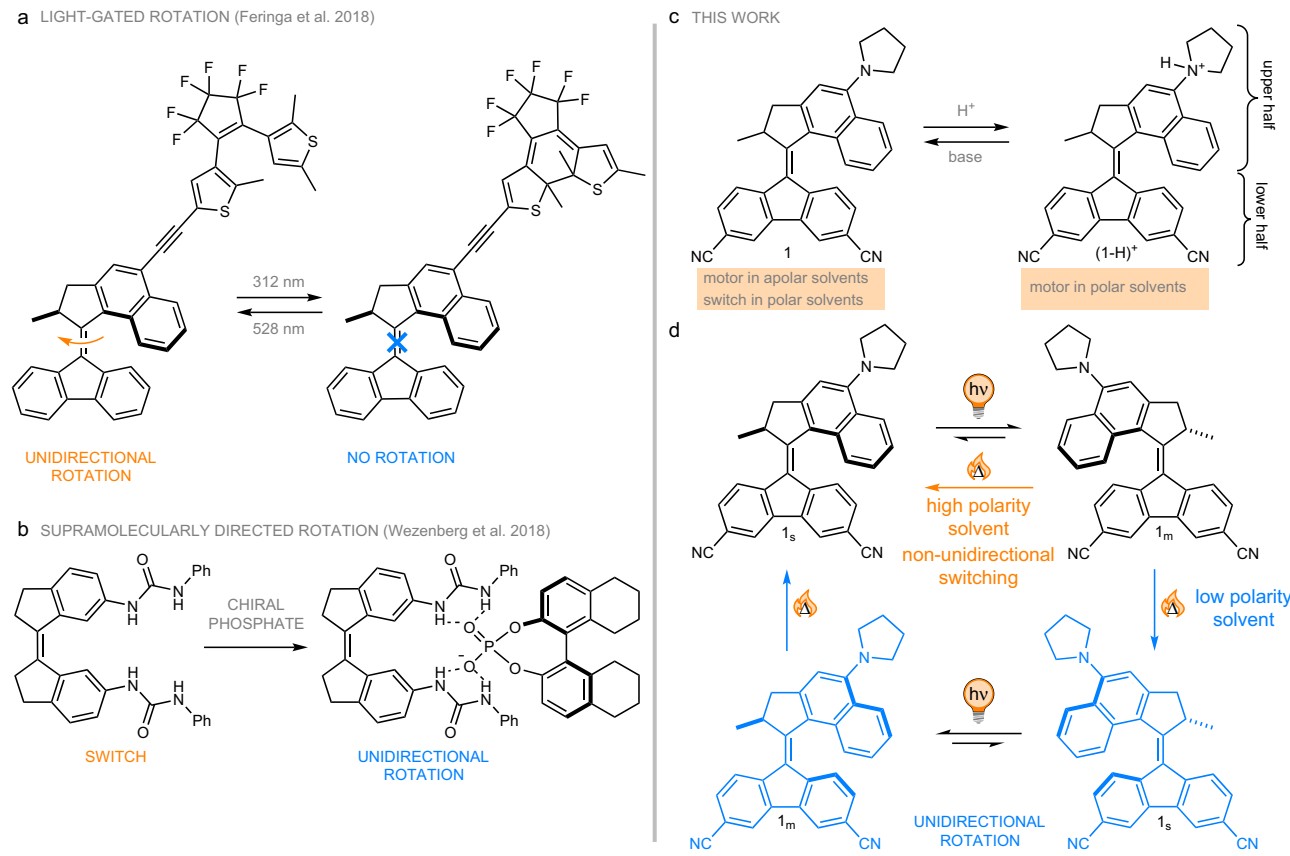

**Fig. 1 Controlling molecular rotation by external stimuli and overview over our current work. a** A light-gated 2nd generation Feringa type molecular motor[54]. **b** A molecular motor prepared from a molecular switch using a supramolecular interaction with a chiral phosphate anion[55,56]. **c** Summary of this work. **d** Isomerization pathways of **1** in polar and apolar solvents.

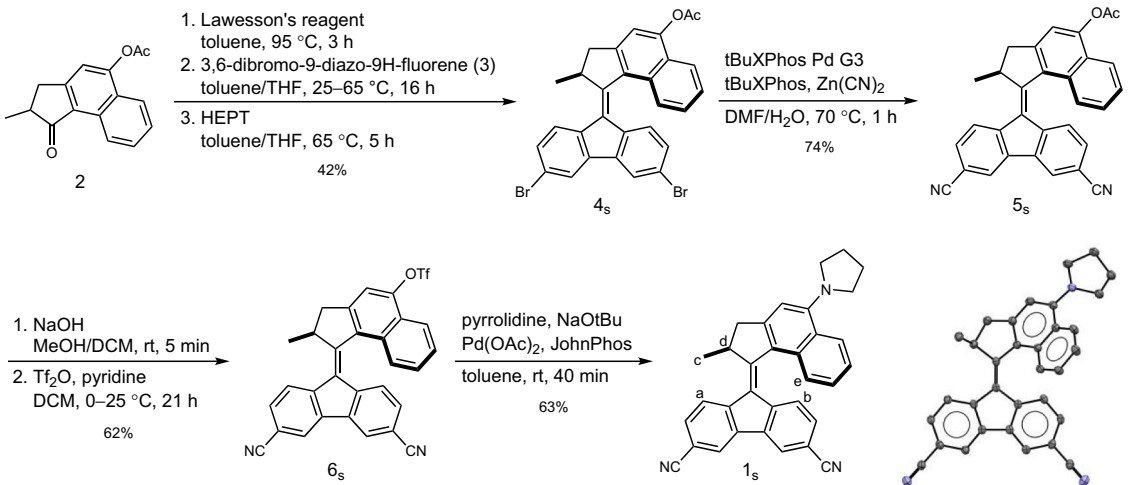

**Fig. 2 Synthesis of 1ₛ shown from the Barton-Kellogg Olefination.** The displacement ellipsoid plot of the single-crystal X-ray structure of **1ₛ** is drawn at 50% probability and hydrogens are omitted for clarity (CCDC 2143227).

The strong and red-shifted absorption of **1ₛ** (Fig. 3a upper panel, Supplementary Fig. 23) compared to the unsubstituted motor core which had been reported previously[59] is attributed to the push-pull nature of the compound, reminiscent of the effect observed in a series of related motors[60]. The presence of both electron donating and accepting moieties can also lead to the formation of the twisted instead of folded structure of overcrowded alkenes concomitant with charge transfer[61]. The twisted structure of **1**, however, is expected not to be able to operate as a rotary molecular motor. These essential structural features were therefore investigated by single-crystal X-ray diffraction as well as 2D NOE NMR (NOESY) experiments. Single-crystals of the as-prepared compound **1ₛ** were grown by vapour diffusion of pentane into a saturated solution of **1ₛ** in DCM resulting in the formation of dark-red needles. The solid-state structure of **1ₛ** in these single crystals as obtained by X-ray diffraction (Fig. 2, Supplementary Fig. 1) is consistent with the folded structure found for stable isomers of overcrowded alkene-based 2nd

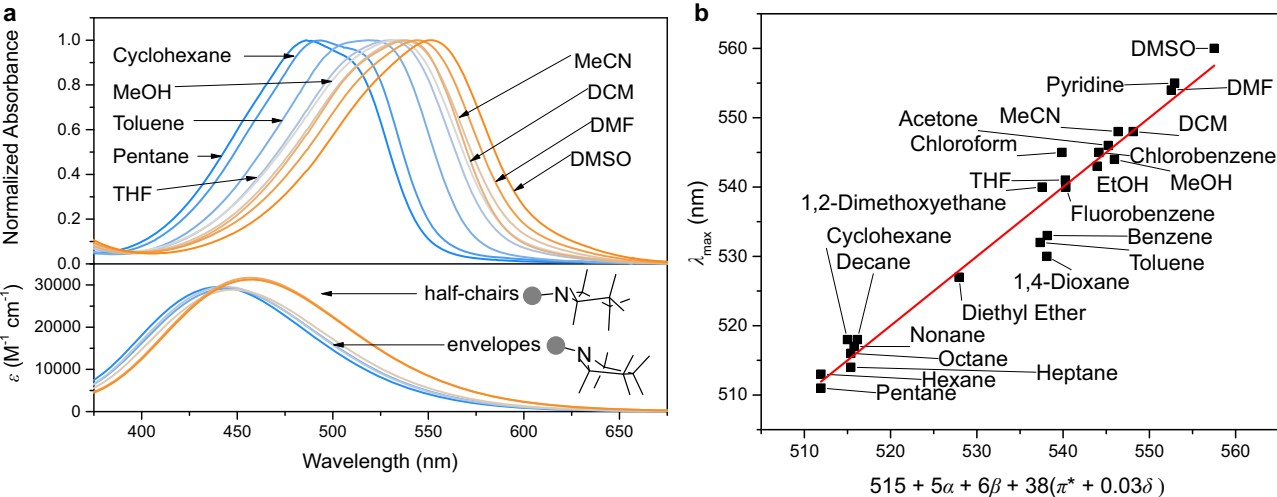

**Fig. 3 Influence of solvent and conformation on absorption properties of 1ₛ. a** Upper panel: Normalized UV-vis absorption spectra of **1ₛ** in different solvents. Lower panel: Gas-phase UV-vis absorption spectra of **1ₛ** obtained by TD-DFT (PBE0/6-311 G+(2d,p)) with the pyrrolidine substituent adopting different envelope (two) and half-chair (four) conformations. **b** Kamlet-Taft plot of **1ₛ** showing the correlation between the absorption maximum ($\lambda_{max}$) of the envelope conformers and the hydrogen bond donor ($\alpha$) and hydrogen bond acceptor ($\beta$) potential as well as the polarizability ($\pi^*$) of the solvent it is dissolved in. A correction term for the presence of halogenated or aromatic solvents ($\delta$) is also considered.

generation rotary molecular motors[62]. A NOESY spectrum of **1ₛ** in CDCl₃ (Supplementary Fig. 3) showed cross peaks for protons $H^a$ and $H^b$ in the lower with $H^c$ and $H^d$ as well as $H^e$ in the upper half (Fig. 2), respectively. This confirms the preservation of the folded structure and therefore the functional, stable isomer **1ₛ** also in solution. No significant differences in the ¹H NMR spectrum of **1ₛ** were found in different deuterated solvents including CDCl₃, toluene-$d_8$, CD₂Cl₂ and MeCN-$d_3$ demonstrating the presence of folded **1ₛ** in solvents with high as well as low dielectric constants ($\varepsilon$).

In short, 2ⁿᵈ generation Feringa type motor **1ₛ** was prepared from commercial starting materials in 8 linear steps with an overall yield of 3.9% and its correct 3D structure was confirmed by X-ray diffraction as well as 2D NMR experiments. During the aforementioned NMR experiments we also observed a positive solvatochromism for **1ₛ**, clearly visible to the naked eye, which will be discussed in the next part.

**UV-vis absorption properties**. Push–pull substituted molecules featuring extended conjugated systems connecting the electron donating and withdrawing moieties often display solvatochromic behaviour as a result of charge transfer transitions in their UV-vis spectra[63–65]. Following our initial observations we collected a series of UV-vis absorption spectra of **1ₛ** in different solvents (Fig. 3a upper panel, Supplementary Fig. 23).

In each solvent the most red-shifted absorption band consisted of two components with the red-shifted contribution increasing relative to the blue-shifted one upon going from low- to high-polarity solvents. We attributed this effect to the presence of varying mixtures of conformers of **1ₛ** with regard to different envelope and half-chair structures of the pyrrolidino substituent. DFT calculations were used to screen the conformational space of the pyrrolidino substituent which revealed four envelope and two half-chair conformations. Optimization of the different structures and prediction of their absorption spectra using time-dependent DFT (TD-DFT, both in gas and in acetonitrile) confirmed our hypothesis that the two components observed in the UV-vis spectra belong to the two conformer types. The envelope conformers give rise to blue-shifted UV-vis absorption spectra compared to the half-chair ones (Fig. 3a lower panel). The role of the increased conjugation of the pyrrolidine nitrogen lone pair

with the motor core for the half-chair compared to the envelope structures emerged by studying the structures and the energy of the conformers in more detail with the aid of Natural Bond Orbital analysis (for details, see SI). The more efficient push–pull effect explains the red-shifted UV-vis absorption spectrum of the half-chair conformers. Moreover, the half-chair structures lead to a more pronounced charge transfer character of the molecule in the ground state, which is evident from the relative energies of half-chair and envelope conformers, with the former becoming more prominent in polar solvents, thereby explaining the change in relative intensities of the two contributions to the lowest-energy electronic transition (S₀→S₁) of **1ₛ** observed experimentally. These spectra show absorption maxima ($\lambda_{max}$) values of the most red-shifted absorption band varying from 511 nm in pentane to 560 nm in DMSO for the contribution corresponding to the half-chair conformers (envelopes: 475 nm in pentane, 532 nm in DMSO), thereby confirming the strong solvatochromic behaviour of **1ₛ**.

After extracting the $\lambda_{max}$ of **1ₛ** in all the solvents used in this study multivariate regression analysis using the Kamlet-Taft solvent parameters[63] was performed for both, half-chair and envelope, conformers using Eq. (1) (Fig. 3b, Supplementary Fig. 32). For this it was assumed that all variants within one conformer type possess identical UV-vis absorption spectra (see ESI Section 6).

$$\lambda_{max} = \lambda_{max,0} + a\alpha + b\beta + c(\pi^* + d\delta) \qquad (1)$$

where $\alpha$, $\beta$ and $\pi^*$ are the solvent parameters for a solvent's hydrogen bond donating and accepting potential as well as dipolarity/polarizability, respectively. $\delta$ is a correction term for the presence of aromatic ($\delta = 1.0$) or halogenated ($\delta = 0.5$) solvents. $\lambda_{max,0}$ is the absorption maximum of the compound in the most apolar solvent measured. Multivariate regression shows a strong correlation between $\lambda_{max}$ of **1ₛ** and $\pi^*$ and smaller contributions by $\alpha$ and $\beta$, for both envelopes and half-chair conformers (see SI). This result can be rationalised by a better stabilisation of charges on the motor in more polar solvents leading to a higher degree of charge transfer between its two halves in all conformers which in turn leads to a redshift of the absorption.

In summary, **1ₛ** shows strong positive solvatochromism for both, half-chair as well as envelope conformations, linked to the

strong push-pull character of its substituents. This is further accentuated by the fact that the half-chair conformations, which possess more red-shifted UV-vis absorption spectra, are present in larger quantities in more polar solvents while the various envelope conformations prefer less polar solvents.

**Isomerization studies on 1.** Next, we set out to study the isomerization behaviour of **1** using UV-vis and NMR spectroscopy. Given the strong influence of solvents' polarity on the absorption properties of **1ₛ**, which we had attributed to varying amounts of charge transfer, we were also keen to study any possible differences in the isomerization behaviour of **1** in solvents of different polarity. For these experiments we, therefore, chose cyclohexane ($\varepsilon = 2.02$) and MeCN ($\varepsilon = 37.5$) as exemplary low and high polarity solvents, respectively.

Figure 4a shows the UV-vis absorption spectrum of **1ₛ** ($1.0 \cdot 10^{-5}$ M) dissolved in cyclohexane at room temperature before irradiation using a 528 nm LED (black line) and after irradiation to the photostationary state (PSS) of **1ₛ** and the metastable isomer **1ₘ** (red line), i.e., the state where upon prolonged exposure no further change was observed. Upon irradiation one can clearly see a redshift of the lowest-energy transition with a new maximum at 522 nm (494 nm for **1ₛ**) which is typical for the formation of the metastable isomers of fluorene-based 2nd generation Feringa type rotary motors[59,66]. The presence of a clear isosbestic point at 521 nm (Supplementary Fig. 22) confirms selective unimolecular E/Z isomerization of the studied motor. The band of the lowest-energy transition in the PSS₅₂₈ₙₘ spectrum also contains multiple new components, hinting at the fact that different conformers of **1ₘ** also give rise to different UV-vis absorption spectra, similar to **1ₛ**. After stopping the irradiation, the PSS₅₂₈ₙₘ spectrum reverted to the original spectrum of **1ₛ** over the course of a couple of hours at room temperature with a clear isosbestic point at 521 nm (Supplementary Fig. 22), identical to the one observed during irradiation. This confirms back-isomerization of **1ₘ** to **1ₛ** following a thermal process. These two consecutive processes of photo-isomerization followed by thermal relaxation can be repeated ten times at room temperature using a 528 m LED without any signs of fatigue of **1** (inset in Fig. 4a).

The blue spectrum in Fig. 4a corresponds to the PSS obtained using a 595 nm LED. This spectrum demonstrates that the motor remains capable of undergoing photochemical isomerization even at this wavelength located near the onset of the lowest-energy transition band. To the best of our knowledge, this is also the longest wavelength reported to date capable of initiating the rotation of a Feringa type rotary motor via one-photon excitation.

By following the thermal back-reaction over time at different temperatures (Eyring analysis) we were able to determine a Gibbs energy of activation $\Delta G^{\ddagger}(20\,°C) = 90.7 \pm 0.1$ kJ·mol⁻¹ in line with the THI of structurally related motors (Supplementary Fig. 24, Supplementary Table 4)[60]. Calculation of the Boltzmann-averaged barrier for THI in the gas phase with DFT leads to a value of 96.0 kJ·mol⁻¹ at the CAM-B3LYP/6-31 G(d,p) level of theory. On the other hand, the barrier for thermal E/Z isomerization was calculated as 107.4 kJ·mol⁻¹, significantly higher than the experimental value. These observations together confirm that **1ₘ** undergoes THI and not thermal E/Z isomerization to regenerate **1ₛ** after photoisomerization in cyclohexane. The isomerization behaviour of **1** in apolar solvents was also studied by ¹H NMR (Fig. 4b). For this, toluene-$d_8$ was used as the solvent due to the better solubility of **1** upon cooling to the temperatures required for this study ($-59\,°C$) in order to quantitatively determine the amount of **1ₘ** present at PSS. Irradiating a sample of **1ₛ** ($2.5 \cdot 10^{-3}$ M) with 530 nm light gave rise

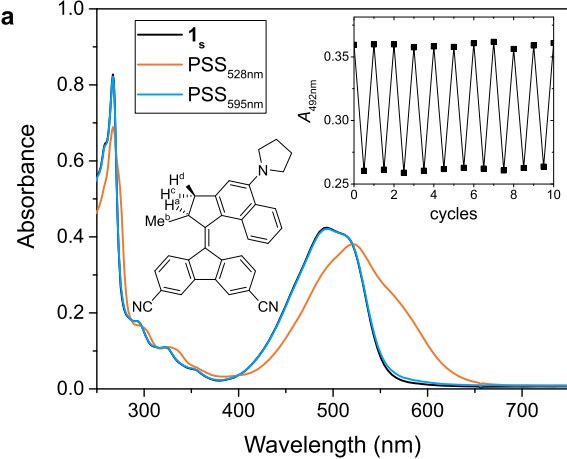

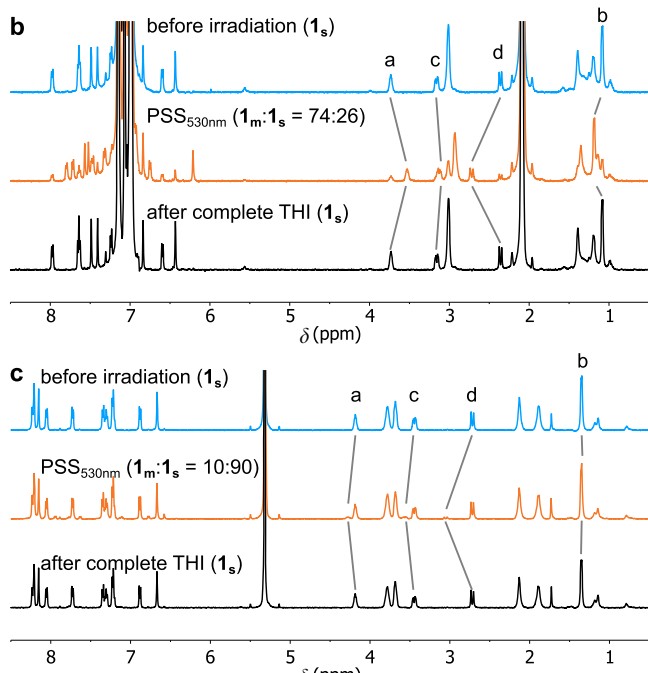

**Fig. 4 Isomerization behaviour of 1 in apolar and polar solvents. a** UV-vis absorption spectra of **1** ($1.0\cdot10^{-5}$ M) in cyclohexane at 20 °C before and after irradiation to PSS using LEDs of different wavelength. The inset shows a fatigue study where the absorbance at 492 nm was followed over ten subsequent 180° rotations. **b** + **c** ¹H NMR spectra of **1** ($2.5\cdot10^{-3}$ M) before and after irradiation to PSS as well as after completed THI (**b**: toluene-$d_8$, $-59\,°C$, 530 nm LED; **c**: CD₂Cl₂, $-85\,°C$, 470 nm LED).

to a new set of signals corresponding to **1ₘ**. The PSS ratio of **1ₘ**:**1ₛ** was determined as 74:26, similar to structurally related 2nd generation motors[60]. The downfield shift observed for methyl Hᵇ typically occurs in Feringa type motors when this group changes from being in a pseudo-axial position (stable isomer) to a pseudo-equatorial one (metastable isomer). Upon warming the sample to room temperature, the original spectrum of **1ₛ** was recovered after a few hours following complete thermal helix inversion.

In contrast to the previous experiments in cyclohexane, irradiation of a sample of **1ₛ** ($1.0 \cdot 10^{-5}$ M) in MeCN at room temperature with a 528 nm LED did not result in changes in the steady-state UV-vis absorption spectrum (Supplementary Fig. 21c). Similar observations were made when using DCM as solvent instead of MeCN. (Supplementary Fig. 21d). Nanosecond

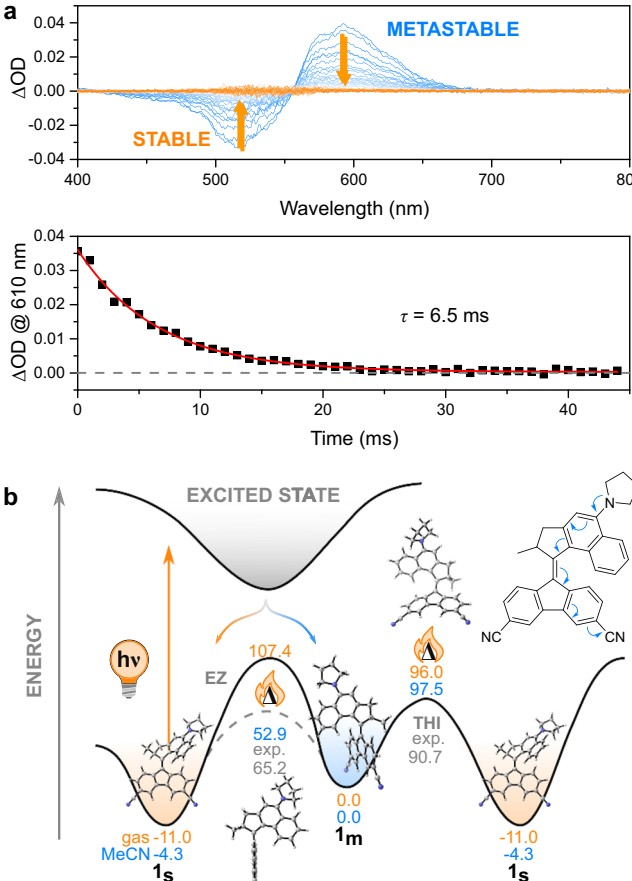

**Fig. 5 Formation of $1_m$ under high polarity conditions and potential energy diagram of 1. a** Top: Thermal recovery of the transient UV-vis signal recorded with ns transient flash photolysis on a sample of $1_s$ dissolved in MeCN. bottom: Decay of the signal at 610 nm attributed to the metastable species. **b** Simplified potential energy diagram for the isomerization pathways of **1** under high (blue) and low polarity (orange) conditions. The Boltzmann-averaged energies taking into account all conformers of the different isomers as obtained by DFT (CAM-B3LYP/6-31 G(d,p)) are reported in kJ·mol$^{-1}$. The solvent contributions were modelled applying the SMD implicit solvation scheme. The structures of **1** represented in the figure belong to the lowest energy pyrrolidine conformer optimised in gas-phase at the CAM-B3LYP/6-31 G(d,p) level. Where available experimental values are shown for comparison.

laser flash photolysis of the sample in MeCN showed, however, that under such conditions a red-shifted transient signal is generated (lifetime of 6.5 ms, Fig. 5a) that we attribute to the metastable form. The better solubility of **1** in $CD_2Cl_2$ compared to MeCN-$d_3$, especially at low temperatures, allowed us to perform low-temperature NMR experiments using this solvent. Such experiments showed that when irradiating a sample of $1_s$ in $CD_2Cl_2$ at −85 °C a set of a weak new signals can be observed by $^1$H NMR that correspond to the metastable isomer $1_m$ (Fig. 4c). The ratio of $1_m$:$1_s$ at PSS was found to be 10:90 and a quick thermal back-reaction to provide the original spectrum of $1_s$ was observed upon warming to room temperature.

These results confirm that $1_s$ still undergoes photochemical E/Z isomerization. However, our experiments indicate that the barrier for the thermal reaction depopulating the metastable state is significantly lower in DCM and MeCN than in cyclohexane, which, in turn, suggests a different depopulation mechanism in these solvents. Recording $^1$H NMR spectra at high magnetic field

strength (≥500 MHz $^1$H NMR frequency) revealed small amounts of $1_m$ to be present in a sample of as-prepared $1_s$ in $CD_2Cl_2$ or MeCN-$d_3$ giving a further indication for a low thermal barrier of interconversion between these two isomers. This allowed us to determine the Gibbs energy of activation for the formation of $1_m$ from $1_s$ as $65 \pm 2$ kJ·mol$^{-1}$ using the Exchange Spectroscopy (EXSY) technique (see ESI Section 4.5). The difference in $\Delta G^{\ddagger}$(20 °C) of ≥20 kJ·mol$^{-1}$ between DCM and cyclohexane is far larger than the ones typically observed for THI of structurally related Feringa type motors[67]. DFT calculations at both the CAM-B3LYP/6-31 G(d,p) and ωB97X-D/def2-SVP levels of theory predict a lower barrier for THI than for thermal E/Z isomerization (96 vs. 107 kJ·mol$^{-1}$ for CAM-B3LYP and 100 vs. 105 kJ·mol$^{-1}$ for ωB97X-D) with a low polarity of the medium ($\varepsilon = 1.0$, approximated in the gas phase), hinting that indeed a change in isomerization mechanism is the reason for the observed difference in $\Delta G^{\ddagger}$(20 °C) for the thermal back-reaction converting $1_m$ to $1_s$.

Figure 5b displays the ground state energy landscape of **1** in low (gas phase, $\varepsilon = 1.0$) and high (MeCN, $\varepsilon = 37.5$) polarity conditions normalized to the stable isomer $1_s$. While an increased polarity has limited influence on the relative energy of the metastable isomer as well as the transition state (TS) of THI, it significantly reduces the barrier for thermal E/Z isomerization via $TS_{EZ}$. In the case of THI, relaxation back to $1_s$ involves the naphthalene moiety of the upper half sliding past the fluorene lower half (Fig. 5b), which leads to considerable steric clash while preserving electronic conjugation between the two motor halves. On the other hand, the TS of the thermal E/Z isomerization shows a perpendicular arrangement of the two motor halves (Fig. 5b), minimizing steric clash but breaking the central alkene and disrupting electronic conjugation. Therefore, a stronger double bond character of the central alkene in $1_m$ and the concomitantly weaker contribution from the molecule's charge transfer state leads to thermal relaxation via helix inversion via $TS_{THI}$ while increased polarity favours E/Z isomerization via $TS_{EZ}$. The latter isomerization can thereby only occur in a backward direction due to the steric clash between upper and lower half of the motor. Similar to the solvatochromism of $1_s$, this unusual behaviour is due to the different amount of charge transfer character of $1_m$ in different solvents weakening the central double bond axle in more polar solvents to a degree where the thermal E/Z isomerization pathway becomes energetically favourable over THI. This behaviour is especially evident analysing the second order orbital interaction energy $E^{(2)}$ obtained from the Natural Bond Orbital Analysis associated with the n→π* donation of the nitrogen atom to the upper half of the molecule. For the most abundant conformation of $1_s$ the increase in polarity is mirrored by $E^{(2)}$ changing from 224 to 278 kJ·mol$^{-1}$ and the Ar–N bond length decreasing by ca. 0.02 Å (from 1.38 to 1.36 Å). An increased effect of polarity was predicted for $1_m$, where the second order perturbation energy increased in polar solvents from 239 to 315 kJ·mol$^{-1}$, accompanied by changes in the Ar–N (1.38 vs 1.36 Å) and C=C (1.38 vs 1.39 Å) bond lengths.

By changing the polarity of the environment around **1**, one can in fact toggle its behaviour between that of a 2$^{nd}$ generation Feringa type rotary motor (apolar solvents) and a molecular switch (polar solvents, see Fig. 1d), something which has never been demonstrated before. This feature is especially of interest for potential applications of motors in environments with changing local conditions, as for example in cells, where it is crucial that the motor function correlates with the environmental parameters. We also envision applications as structural components in dynamic porous solids where filling the pores with solvents of different polarity could be used to change their properties in situ.

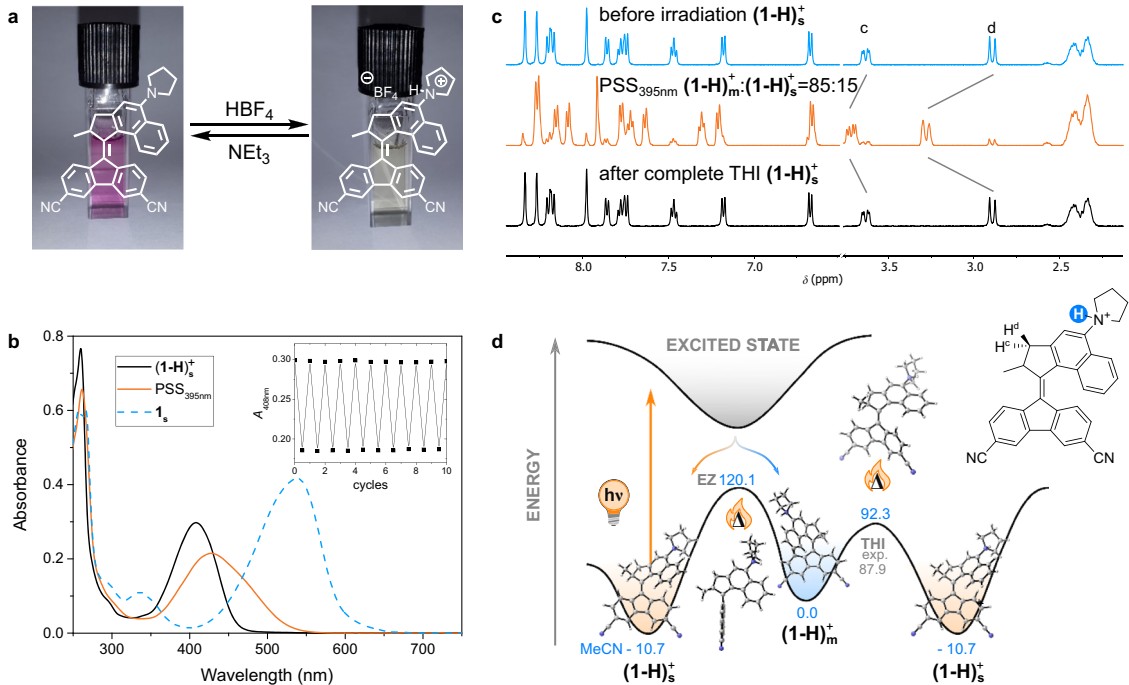

**Fig. 6 Formation and isomerization behaviour of (1–H)⁺. a** Interconversion of **1ₛ** and **(1–H)ₛ⁺. b** UV-vis spectra of **(1–H)⁺** (1.0·10⁻⁵ M) in MeCN at 20 °C before and after irradiation to PSS with a 395 nm LED. The spectrum of **1ₛ** before addition of HBF₄·OEt₂ is shown for comparison. The inset shows a fatigue study where the absorbance at 408 nm was followed over ten subsequent 180° rotations. **c** ¹H NMR spectra **(1–H)⁺** (2.5·10⁻³ M) in MeCN-d₃ at −30 °C before and after irradiation to PSS with a 395 nm LED as well as after completed THI. **d** Potential energy diagram of isomerization pathways of **(1–H)⁺** under high polarity conditions. The energies of the lowest energy pyrrolidine conformer as obtained by DFT (SMD(MeCN)-CAM-B3LYP/6-31 G(d,p)) are reported in kJ·mol⁻¹. The experimental value for the activation barrier of THI is shown for comparison.

**Formation of and isomerization studies on** (1–H)⁺. Given our observation that the ability of our compound to alternate its behaviour between that of a unidirectional motor and that of a molecular switch in response to environmental factors is due to its push–pull character, we speculated that by temporarily blocking this feature the motor function could be restored in polar solvents, giving us an additional chemical handle to control its mechanical properties.

In order to eliminate the push–pull character of **1**, protonation of the pyrrolidino nitrogen is ideal, as (i) it does not require synthesis of a new compound, (ii) can be performed in situ and (iii) is easily reversible. By doing so we are changing a strongly electron donating substituent ($\sigma_p$(NMe₂) = −0.83)[68] to a strongly electron withdrawing one ($\sigma_p$(N⁺Me₃ = 0.82)[68]. We therefore titrated a solution of **1ₛ** (1.0 · 10⁻⁵ M) in MeCN with a 1.0 v/v% solution of HBF₄ · OEt₂ in MeCN recording a UV-vis absorption spectrum at the beginning and after each addition (Supplementary Fig. 26a). Over the course of this addition the lowest-energy transition maximum at 539 nm disappeared and a new band formed around 406 nm corresponding to the protonated stable isomer **(1–H)ₛ⁺** (Fig. 6a, b). This new $\lambda_{max}$ is slightly more red-shifted compared to the unsubstituted parent motor ($\Delta\lambda_{max}$ = 16 nm), most likely due to the extension of the π-electron system by the CN groups in the lower motor half[60]. A clear isosbestic point was observed for this protonation confirming a clean unimolecular process. The new band also shows a Gaussian shape without a shoulder as is the case for unprotonated **1ₛ**. This indicates that the different conformations of the protonated pyrrolidino substituent no longer have a significant impact on the absorption properties of this motor. Adding 1.0 eq. (relative to HBF₄ · OEt₂) of NEt₃ to a solution of **(1–H)ₛ⁺** leads to the recovery of the UV-vis absorption spectrum of **1ₛ**. This process of protonation-deprotonation could also be followed by ¹H NMR

(Supplementary Figs. 12 and 13) and could be repeated three times without significant signs of fatigue (Supplementary Fig. 26b). Comparing the spectra of **(1–H)ₛ⁺** in different solvents furthermore revealed a negligible solvatochromic effect, confirming that this effect was a result of the push–pull character of **1ₛ** (Supplementary Fig. 29).

Irradiating a solution of **(1–H)ₛ⁺** (1.0 · 10⁻⁵ M) in MeCN with a 395 nm LED until no further spectral changes were observable (PSS) resulted in a redshift of the lowest-energy transition band in the UV-vis spectrum characteristic for the formation of the metastable isomer **(1–H)ₘ⁺** (Fig. 6b). After removal of the light source the spectrum slowly returned to that of **(1–H)ₛ⁺** over the course of one hour. For both processes a clear isosbestic point was observed at 430 nm (Supplementary Fig. 28). Following these isomerizations by ¹H NMR using a more concentrated solution of **(1–H)ₛ⁺** (2.5 · 10⁻³ M) in MeCN-d₃ provided the photostationary distribution of **(1–H)ₘ⁺:(1–H)ₛ⁺** at the PSS as 85:15 and gave further proof of the excellent selectivity of these processes (Fig. 6c). Eyring analysis of the thermal back-reaction allowed us to determine $\Delta G^{\ddagger}$(20 °C) as 87.9 ± 0.2 kJ·mol⁻¹ in MeCN (Supplementary Fig. 30, Supplementary Table 5), very similar to the 90.7 ± 0.1 kJ·mol⁻¹ of **1ₘ** in cyclohexane (Supplementary Fig. 24, Supplementary Table 4) which we had assigned to THI rather than thermal E/Z isomerization using DFT calculations. In the present case, these calculations at the (SMD)-CAM-B3LYP/6-31 G(d) level of theory predicted barriers of 92.3 kJ·mol⁻¹ for THI and 120.1 kJ·mol⁻¹ for thermal E/Z isomerization of **(1–H)ₘ⁺** in MeCN (SMD solvent model). Similar bond lengths of the central alkene were found in the DFT optimized structures of **1ₘ** and **(1–H)ₘ⁺** (1.39 vs 1.38 Å) rationalizing the almost identical activation barriers for THI which are governed by the steric clash in the so-called fjord region between upper and lower motor half. This result confirms that upon protonation of **1ₛ** its motor

function in polar solvents can be fully restored providing a second means to switch the behaviour of this photoactive molecule between that of a motor (protonated) and a switch (unprotonated). The potential energy diagram of a 180° rotation of $(1–H)^+$ in polar solvents therefore looks as shown in Fig. 6d. Comparable behaviour to the one described here for MeCN was also found for DCM (see ESI for details).

Using a combination of UV-vis absorption and NMR experiments backed up by DFT calculations we have shown that apart from controlling the light absorption properties of compound **1** also its mode of operation can be toggled between that of a 2$^{nd}$ generation rotary molecular motor and a photochemical switch using external stimuli. This unusual and unprecedented behaviour is rooted in the strong push-pull character of **1** making it highly sensitive to changes in the polarity of its environment. Following photoisomerization from $\mathbf{1_s}$ to $\mathbf{1_m}$, this metastable isomer can revert to the global minimum structure ($\mathbf{1_s}$) via two possible pathways, forward motion via THI or backward motion via thermal $E/Z$ isomerization. While the energetic barrier for THI is mainly guided by the steric clash in the so-called fjord region between the naphthalene and fluorene moieties of the upper and lower motor half, respectively, the dominant factor in the case of thermal $E/Z$ isomerization is the loss of electronic conjugation between the two motor halves with concomitant formation of a biradical or charge transfer TS. Our DFT calculations have shown that the barrier for THI is not sensitive to polarity as the steric clash in $TS_{THI}$ remains almost unchanged. On the other hand, increased stabilization of structures with charge transfer character under more polar conditions significantly reduces the energy of $TS_{EZ}$ compared to $\mathbf{1_m}$. Consequently, under low polarity conditions **1** operates as a unidirectional rotary molecular motor, similar to structurally related motors, due to the lower energetic barrier being associated with the THI pathway. Increasing the polarity, however, makes thermal $E/Z$ isomerization the preferred pathway of relaxation. Additionally, we discovered that by protonating the pyrrolidino substituent of compound **1** and thereby cancelling out its push–pull character, **1** returned to operate as a unidirectional motor under high polarity conditions, affording an orthogonal mean for controlling its dynamic behaviour.

In short, we have introduced a concept to control the mode of operation of a nanosized molecular machine between that of a unidirectional motor with exclusive forward rotary motion and a back-and-forth molecular switch using solely external stimuli. The fact that no prior synthetic alteration of these compounds is necessary makes this a highly attractive approach for applications requiring in situ control over molecular motion. We therefore expect that our findings will create many new and exciting opportunities to control molecular machine behaviour and allow more complex mechanical functions in the field of smart and responsive materials using orthogonal external stimuli.

## Methods

**Steady-state UV-vis absorption spectroscopy.** $1.0 \cdot 10^{-5}$ M solutions of $\mathbf{1_s}$ were prepared in the specified solvents. For each experiment 2.5 mL of the according sample was transferred into a 1.0 cm quartz cuvette. For protonation an excess of $HBF_4 \cdot OEt_2$ was added inside a glove box. Samples were placed in the cuvette holder of an Agilent 8453 UV-vis Diode Array System, equipped with a Quantum Northwest Peltier controller, and left to equilibrate at the indicated temperature before recording the first spectrum. Irradiation experiments were performed using fiber-coupled LEDs (M395F3, M595F2) obtained from Thorlabs Inc. as well as a home-made system using a 528 nm LED (OSRAM Oslon SSL 80 LTCP7P-KXKZ (KZ)).

**NMR spectroscopy.** For isomerization studies, $2.5 \cdot 10^{-3}$ M solutions of $\mathbf{1_s}$ were prepared in the specified solvents. For each experiment 500 µL were transferred into an NMR tube. For studies requiring $(1–H)^+$ 1.0 µL of $HBF_4 \cdot OEt_2$ was added. The NMR tube was subsequently fitted with a glass fiber cable for in situ irradiation

and the samples were left to equilibrate at the specified temperature. To follow the protonation of $\mathbf{1_s}$ $1.3 \cdot 10^{-3}$ M solutions were used. NMR spectra were recorded on a Varian Unity Plus 500 NMR spectrometer. Chemical shifts are given in parts per million (ppm) relative to the residual solvent signal. Irradiations were performed using fiber-coupled LEDs (M395F3, M470F3, M530F2) obtained from Thorlabs Inc.

**Transient absorption spectroscopy.** Nanosecond transient absorption spectra were recorded with an in-house assembled setup. To generate the excitation wavelength of 510 nm a tunable Nd:YAG-laser system (NT342B, Ekspla) consisting of the pump laser (NL300) with harmonics generators (SHG, THG) producing 355 nm light to pump an optical parametric oscillator (OPO) with SHG connected in a single device was used. The laser system was operated with a pulse length of 5 ns and a repetition rate of 10 Hz. The probe light running at 20 Hz was generated by a high-stability short arc xenon flash lamp (FX-1160, Excelitas Technologies) employing a modified PS302 controller (EG&G). The probe light was split equally into a signal and a reference beam by means of a 50/50 beam splitter and focused (bi-convex lens 75 mm) on the entrance slit of a spectrograph (SpectraPro-150, Princeton Instruments) with a grating of 150 ln·mm$^{-1}$, blaze at 500 nm. The probe beam ($A = 1$ mm$^2$) was set to pass through the sample cell and orthogonally overlapped with the excitation beam on a 1 mm × 1 cm area. The excitation energy was determined by recording the excitation power at the back of an empty sample holder. To correct for fluctuations in the spectral intensity of the flash lamp, the signal was normalized using the reference. The two beams were recorded at the same time using a gated intensified CCD camera (PI-MAX3, Princeton Instruments) with an adjustable gate of minimal 2.9 ns. Under standard settings a gate of 20 ns and software binning was used to improve the dynamic range and signal to noise ratio. Two delay generators (DG535 and DG645, Stanford Research Systems, Inc.) were used to trigger the excitation and to change the delay of the flash lamp together with the gate of the camera during the experiment. An in-house written Labview program was used to control the setup.

**Computational analysis.** All calculations for the thermal isomerization of **1** and for its protonated analogue $(1–H)^+$ were carried out using Gaussian 16 Rev. B.01[69]. The mechanisms were analyzed at the CAM-B3LYP/6-31 G(d,p) level and the results were compared to the ones obtained at the ωB97X-D/def2-SVP level. The SMD implicit solvent model was used to model acetonitrile ($\varepsilon = 37.5$) solvation. A broken-symmetry approach was included to take into account the possible diradical nature of the $E/Z$ isomerization transition state. In all cases the closed-shell solution was retrieved. All the optimizations were confirmed to be stationary points by the number of imaginary frequencies (0 for the minima, 1 for the transition state). The monodeterminantal nature of the transition state (due to the marked charge transfer character) was confirmed by means of a minimal CASSCF(10,10)/6-31 G* single point calculation using OpenMolcas v18.09[70] on $TS_{EZ}$ optimized in gas phase at the CAM-B3LYP/6-31 G(d,p) level. The closed shell conformation represented >89% of the ones possible for the transition state. Quantitative treatment of the nitrogen lone-pair donation was obtained through NBO analysis at the CAM-B3LYP/6-31 G(d,p) level with the NBO6 software package[71]. The cartesian coordinates of the compounds studied in this work are reported in the file Supplementary Data 1.

## Data availability

All data supporting the findings presented here are presented in the accompanying Supporting Information. Crystallographic data have been deposited with the Cambridge Crystallographic Data Centre (CCDC 2143227) and can be obtained free of charge. Other related data are available from the corresponding author upon request.

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

## Acknowledgements

The authors are grateful for financial support from the Netherlands Organization for Scientific Research (NWO-CW), the European Research Council (ERC; advanced grant No. 694345 to B.L.F.), the Dutch Ministry of Education, Culture and Science (Gravitation program No. 024.001.035) and Marie Skłodowska-Curie Actions (Individual Fellowship No. 793082 for L.P. and 838280 to S.C.). Thanks go to the Centre for Information Technology of the University of Groningen for their support and for providing access to the Peregrine high-performance computing cluster.

## Author contributions

Conceptualization: L.P. and B.L.F. Funding: L.P., S.C. and B.L.F. Synthesis and characterization of compounds: L.P. Single-crystal X-ray analysis: L.P. Steady-state UV-vis absorption spectroscopy: L.P. and S.C. NMR experiments: L.P., S.C., P.V.D.M., J.K. and R.M.S. DFT studies: L.P. and S.C. Time-resolved transient absorption studies: S.C., M.F.H. and W.J.B. Project administration: L.P., S.C. and B.L.F. Supervision: B.L.F. Writing – original draft: L.P. and S.C. Writing – review and editing: L.P., S.C., P.V.D.M., J.K., R.M.S., M.F.H., W.J.B. and B.L.F.

## Competing interests

The authors declare no competing interests.
