## [Peer Review File · Nature Communications]

Controlling Forward and Backward Rotary Molecular Motion on DemandREVIEWER COMMENTS

Reviewer #1 (Remarks to the Author):

In this work B. Feringa and coworkers describe a very interesting concept of a molecular motor whose motions can reversibly be toggled between a unidirectional rotary motion and a reversible non-directional back-and-forth switching motion. To this end a second generation “2-step” motor is endowed with a conjugated strong push-pull substitution pattern consisting of two cyano-groups at the stator and a pyrrolidine substituent at the rotor fragment. The effects of the strong push-pull character on the ground state energy barrier for thermal double-bond isomerization is significantly altered in response to the polarity of the solvent. In low polarity solvents thermal helix inversion is significantly faster than thermal double bond isomerization and thus unidirectional motor operation proceeds consisting of photoisomerization of the double bond and subsequent thermal helix inversion. In high polarity solvents the push-pull character is increased so much that double bond isomerization becomes significantly faster than thermal helix inversion, leading to a switch-like behavior: photoisomerization of the double bond to the meta-stable isomer and subsequent thermal back-isomerization of the double bond to the starting point. The behavior in polar solvents can be reverted back to unidirectional motor rotation via protonation of the pyrrolidine substituent, which effectively eliminates its donating capacity and abolishes the push-pull properties in the motor. Simple protonation/deprotonation can therefore be used to change the behavior of this neat molecular machine.

The approach is certainly of high novelty and conceptual originality. The work is done at a very high level both experimentally as well as theoretically, the analysis is comprehensive and well documented. The manuscript is written very clearly and well organized. Overall I can recommend publication in Nature Communications of this fine work after some minor points are addressed as outlined below.

- Although I find the combined experimental and theoretical assignments of the processes convincing I wanted to get the authors opinion/reasoning why a two-step motor was chosen instead of a nonsymmetric four-step motor to evidence the here presented concept. In a four-step motor it would be straight-forward to distinguish experimentally with simple NMR spectroscopy between thermal helix inversion and double bond isomerization steps. This is not well possible without the help of calculations and discussing/arguing ground state energy barriers in a two-step motor.
- The authors discuss to some length the UV-vis spectra shapes and assign them to different conformer populations. Although their calculations are convincing for this assignments, can the authors exclude vibrational contributions as the reasons for the spectral shapes?
- The authors deliver calculated energy barriers for the thermal double bond isomerization. They handle the multi-configurational character at the transition state for this process by a broken symmetry approach. I assume that the strong push-pull character allows the DFT description to be accurate in this case and a zwitter-ionic transition state is observed or is a biradical character dominating?
- “a standard Gibbs free energy of $\Delta G^\ddagger(20\text{ }^\circ\text{C}) = 90.7 \pm 0.1\text{ kJ}\cdot\text{mol}^{-1}$ ” should be Gibbs energy of activation. Also at a later point “Gibbs free energy of formation” is used. This should also be Gibbs energy of activation.
- In the text, experimentally obtained values for THI are discussed, whereas in the figures / energy schemes only values from DFT calculations are shown. This is a bit confusing. Please add both experimental and theoretical values where available.
- In Fig 6d, the transition state for EZ isomerization is assigned an energy of 120.1 kJ/mol. However, in the text the height of the barrier for (1-H)+m to (1-H)+s EZ isomerization is given as 120.1 kJ/mol. As (1-H)+m is assigned a relative energy of 10.7 kJ/mol, one of the values must be off by 10.7 kJ/mol. The same is true for the THI process.
- Fig 1b chiral phosphate (on the arrow) is too small, P-O (double)bonds are not visible
- Fig 1d is difficult to understand 1-H+ should be a motor in polar solvents (Fig 1c), but in 1-d is written: high-polarity solvent, non-unidirectional switching. Better separate protonated and deprotonated structures.
- Fig 2 Layout (reaction conditions) is not consistent, XRAY should have transparent/white

background

- Fig 3a what are the "different conformations"? There are four curves but only two are labelled. Or is this actually an overlay/summary of multiple curves? Maybe label the "envelopes" curves separately.
- Fig 6b 1-H should be (1-H)+
- small number of typos: ("stead-state", "prove" instead of "proof")

Supporting Information

- the 2D spectra are shown without signal assignments, which would help to better understand what is shown
- 2D but also 1D NMR axis should be labelled delta not f1, f2
- p 38 Kamlet Taft analysis: the letter beta looks a bit strange

Reviewer #2 (Remarks to the Author):

This paper develops strategies to control the function of overcrowded-alkene rotary molecular motors. The starting point is a newly synthesized but "ordinary" second-generation motor featuring a push-pull system in the form of one electron-donating amino substituent and two electron-withdrawing cyano substituents. Owing to this push-pull system, it is demonstrated that while the motor maintains its rotary function in apolar solvents, this function is lost in polar solvents, whereby it rather behaves as a molecular switch that does not produce net work upon interconversion between its stable forms. Subsequently, it is shown that the motor function can be regained in polar solvents by extinguishing the push-pull system through protonation of the amino group.

Overall, this is a very nice piece of work where several experimental techniques as well as quantum chemical methods are used to corroborate the findings reported. Furthermore, the presentation is clear and the paper is easy to read. Overall, I only have two minor comments that I think should be addressed in a revised manuscript:

(1) The scenario put forth to explain why the rotary function is lost in polar solvents is that the barrier for the thermal E/Z isomerization is lowered below that of the thermal helix inversion needed to perpetuate the forward rotary motion. However, the thermal E/Z isomerization is much less affected by steric considerations than the thermal helix inversion. Thus, it is not immediately clear to me why the thermal E/Z isomerization would have to occur in a "backward" direction and prevent rotary motion? I think some comments on this issue in a revised manuscript would improve the paper.

(2) From the computational modelling, it seems to me that the thermal helix inversion is predicted to occur in a single step (only a single transition state is located for this process)? However, this is at odds with previous computational studies using similar methods (see, for example, *J. Org. Chem.* 2014, 79, 927 and *Phys. Chem. Chem. Phys.* 2015, 17, 21740).

Reviewer #3 (Remarks to the Author):

In this work Pfeifer and coworkers report on the reversible switching of the behavior a Feringa-type motor (overcrowded alkene) between that of a motor and that of a switching upon changing the polarity of the medium or the protonation state of the molecule.

Pfeifer et al. ascribe this behavior to the push-pull system between the two halves of the motor. A combination of experimental kinetic data and DFT calculations allows them to conclude that the activation energy for thermal E/Z isomerization is highly sensitive to the medium's polarity when in the deprotonated form, due to the push-pull character of the chromophore. Conversely, the activation energy for THI is mostly unaffected. As a result in the deprotonated state the molecule behave as a motor (unidirectional rotation $E_a(\text{THI}) < E_a(\text{E/Z})$) in low polarity media, and as a switch in high polarity media (no unidirectional rotation $E_a(\text{THI}) > E_a(\text{E/Z})$). Deactivation of the push-pull character by

protonation of the tertiary amine restore the motor functionality in polar media.

While, the use of external stimuli to control molecular machines is known, the novelty lays in its use to toggle the energy ratchet behavior of the system.

Conclusions are substantiated by the experimental results and these are clearly explained in the main text. I can thus recommend publication provided that some minor issues are addressed (see below).

1. While the authors make an effort to rationalize the effect of polarity and protonation on the activation energy of the thermal E/Z isomerization, it is never clearly rationalized in the text why the THI activation energy should not be affected by the polarity of the medium or by the presence/absence of the push-pull character. Despite their hypothesis being intuitively and qualitatively correct and their message can be grasped by the reader, I believe that the reader could benefit from a clear (possibly quantitative) statement that rationalize this point. Also considering the high level of the experimental and calculated data.
2. It would be nice to report, along with the enthalpy and entropy of activation, the Eyring plots for the THI process examined by NOESY (ESI section 4.5).
3. In table S4, the temperature of determination is needed only for the ΔG of activation. Enthalpy and entropy are extracted from Eyring analysis under the assumption of being temperature-independent. Therefore the notation $\Delta H(20^\circ\text{C})$ has no meaning in this context.
4. In the caption of the NMR spectra should be reported the field of the instrument as well as the t_{mix} used for acquiring the NOESY spectra. These are particularly important when dealing with dynamic NMR.
5. In figure S10 and S11 signal "a" shifts upfield when irradiating at 530 nm, while it shifts downfield when irradiating at 470 nm. As the product of E/Z photoconversion is expected to be the same, what can this difference be ascribed to? Is this simply due to the solvent in which the spectra are recorded?
6. Consistency should be used in the temperature units between the main text and the SI and within the SI, they are reported sometimes in Kelvin and sometimes Celsius.
7. At line 242 I would add the solvent explicitly, so it reads like "... 1m undergoes THI and not... after photoisomerization in hexane". At line 286 I believe the authors meant the "free energy of activation", which is a kinetic parameter that can be extracted by EXSY, and not the "standard free energy of formation", which is a thermodynamic parameter and to my understanding has no meaning in this context.

SC

Responses to Reviewers' Comments

Reviewer 1

We would like to thank Reviewer 1 for the positive comments on our work and the extraordinarily thorough review of our manuscript and supporting information. We gladly address all the questions and concerns raised below.

Although I find the combined experimental and theoretical assignments of the processes convincing I wanted to get the authors opinion/reasoning why a two-step motor was chosen instead of a nonsymmetric four-step motor to evidence the here presented concept. In a four-step motor it would be straight-forward to distinguish experimentally with simple NMR spectroscopy between thermal helix inversion and double bond isomerization steps. This is not well possible without the help of calculations and discussing/arguing ground state energy barriers in a two-step motor.

The main reason for choosing to work with a motor bearing a symmetrical lower half, making the two stable isomers identical with the same also being true for the metastable isomers, was synthetic accessibility. We found it impossible to conduct a successful Barton-Kellogg coupling using precursors with already installed strong electron donating and withdrawing groups. Therefore, desymmetrization of the lower motor half could have only been carried out with the motor scaffold already in place. This is a highly challenging task for two reasons; the lack of selectivity of the associated reactions as well as the difficult separation of stereoisomers which will be formed as byproducts. As pointed out in the manuscript, the stability of electronically strongly biased overcrowded alkenes also causes problems under some reaction conditions, for example when working with strong bases. This further limits the synthetic possibilities following Barton-Kellogg coupling. Therefore, after finally having found a route to make our motor in suitable quantities, we settled for the approach requiring more analytical rigor.

The authors discuss to some length the UV-vis spectra shapes and assign them to different conformer populations. Although their calculations are convincing for this assignments, can the authors exclude vibrational contributions as the reasons for the spectral shapes?

Vibrational contributions as the reasons for the spectral shapes can be ruled out based on previous results obtained on closely related motors. The unsubstituted parent motor as well as a series of motors bearing different substituents in the upper half showed a perfectly symmetrical band for the lowest energy transition (*Org. Biomol. Chem.* **2008**, *6*, 1605–1612.). The same was true for the two push-pull motors presented in another one of our previous publications (*Chem. Sci.* **2019**, *10*, 8768–8773.). This is also true for the protonated motor presented in the current manuscript, where the different conformations of the pyrrolidino substituent don't affect the electronic structure of the motor's π -electron system due to the lack of a lone pair on the nitrogen. Therefore, the observed spectral features are not a result of the motor backbone, nor the presence of a push-pull system but are due to the unprotonated pyrrolidino substituent.

The authors deliver calculated energy barriers for the thermal double bond isomerization. They handle the multi-configurational character at the transition state for this process by a broken symmetry approach. I

assume that the strong push-pull character allows the DFT description to be accurate in this case and a zwitter-ionic transition state is observed or is a biradical character dominating?

Indeed, the push-pull character dominates in the TS. We modified the Computational Analysis section adding the following sentence: "A broken-symmetry approach was included to take into account the possible diradical nature of the E/Z isomerization transition state. In all cases the closed-shell solution was retrieved."

"a standard Gibbs free energy of $\Delta G^\ddagger(20\text{ }^\circ\text{C}) = 90.7 \pm 0.1\text{ kJ}\cdot\text{mol}^{-1}$ " should be Gibbs energy of activation. Also at a later point "Gibbs free energy of formation" is used. This should also be Gibbs energy of activation.

The text was adapted according to the suggestions of the reviewer. Changes are highlighted in yellow.

In the text, experimentally obtained values for THI are discussed, whereas in the figures / energy schemes only values from DFT calculations are shown. This is a bit confusing. Please add both experimental and theoretical values where available.

Fig. 5 and 6 were changed according to the suggestions of the reviewer.

In Fig 6d, the transition state for EZ isomerization is assigned an energy of 120.1 kJ/mol. However, in the text the height of the barrier for (1-H)+m to (1-H)+s EZ isomerization is given as 120.1 kJ/mol. As (1-H)+m is assigned a relative energy of 10.7 kJ/mol, one of the values must be off by 10.7 kJ/mol. The same is true for the THI process.

We thank the referee for spotting this mistake. We have corrected the energies in the picture, because they were referred to the metastable state and not to the stable state.

Fig 1b chiral phosphate (on the arrow) is too small, P-O (double)bonds are not visible.

Fig. 1b was changed according to the suggestions of the reviewer.

Fig 1d is difficult to understand 1-H+ should be a motor in polar solvents (Fig 1c), but in 1-d is written: high-polarity solvent, non-unidirectional switching. Better separate protonated and deprotonated structures.

We thank the reviewer for pointing this out. To avoid confusion, we decided in Fig 1d to focus the attention only on motor 1, because the concept of motor in polar solvent for the protonated species is already explained in Fig. 1c.

Fig 2 Layout (reaction conditions) is not consistent, XRAY should have transparent/white background.

Fig. 2 was changed according to the suggestions of the reviewer.

Fig 3a what are the “different conformations”? There are four curves but only two are labelled. Or is this actually an overlay/summary of multiple curves? Maybe label the “envelopes” curves separately.

In our calculations we have found six conformers for the stable state 1s, two attributable to half-chair conformers and four to envelopes, as we described in the Supporting Information. We have modified the figure to show more clearly that we are referring to different sets of spectra and modified the caption that now reads “Gas-phase UV-vis absorption spectra of 1s obtained by TD-DFT (PBE0/6-311G+(2d,p)) with the pyrrolidine substituent adopting different envelope (two) and half-chair (four) conformations”

Fig 6b 1-H should be (1-H)+.

Fig. 6 was changed according to the suggestions of the reviewer.

small number of typos: (“stead-state”, “prove” instead of “proof”)

Changes were made accordingly and are highlighted in yellow.

Supporting Information

The 2D spectra are shown without signal assignments, which would help to better understand what is shown.

Signal assignments have been added to the characterization data and 2D NMR spectra of **1_s** and **(1-H)_s⁺**. Changes are highlighted in yellow.

2D but also 1D NMR axis should be labelled delta not f1, f2.

Changed accordingly. Changed spectra are highlighted in yellow.

p 38 Kamlet Taft analysis: the letter beta looks a bit strange.

This point has been addressed. Changes are highlighted in yellow.

Reviewer 2

We thank the reviewer for the positive comments on our manuscript and for taking the time to review and propose ways of improving it. The issues raised by the reviewer have been addressed as follows.

The scenario put forth to explain why the rotary function is lost in polar solvents is that the barrier for the thermal E/Z isomerization is lowered below that of the thermal helix inversion needed to perpetuate the forward rotary motion. However, the thermal E/Z isomerization is much less affected by steric considerations than the thermal helix inversion. Thus, it is not immediately clear to me why the thermal E/Z isomerization would have to occur in a “backward” direction and prevent rotary motion? I think some comments on this issue in a revised manuscript would improve the paper.

We thank the reviewer for raising this question as it lies at the heart of our concept. Indeed, thermal E/Z isomerization from the metastable isomer can only occur in a “backward” direction (i.e. undoing the movement of the preceding photochemical E/Z isomerization) because in the metastable isomer the upper motor half has not yet passed over the lower one. In other words, the naphthalene unit of the upper half still resides on the same side of the fluorene lower half as before the photochemical E/Z isomerization. Therefore, in order for this isomerization to proceed in a forward fashion the barrier due to the steric clash between the two motor halves would have to be overcome. By overcoming this barrier, the motor, however, arrives at the second stable isomer. This process is what we call thermal helix inversion.

We have added the following sentence to the manuscript to clarify this point: The latter isomerization can thereby only occur in a backward direction due to the steric clash between upper and lower half of the motor.

From the computational modelling, it seems to me that the thermal helix inversion is predicted to occur in a single step (only a single transition state is located for this process)? However, this is at odds with previous computational studies using similar methods (see, for example, J. Org. Chem. 2014, 79, 927 and Phys. Chem. Chem. Phys. 2015, 17, 21740).

The two papers mentioned by the reviewer are dealing with 2nd generation molecular motors with a structurally different backbone, where one or both of the 5-membered rings next to the central alkene are replaced with 6-membered ones. This has profound influences on the mechanism of thermal helix inversion such that in the case of two 5-membered rings this thermal reaction occurs in a single step. This has also been described in previous publications (e.g. J. Phys. Chem. A **2010**, 114, 5058–5067).

Reviewer 3

We thank the reviewer for the positive comments on our manuscript. The points that were raised have all been addressed as follows.

While the authors make an effort to rationalize the effect of polarity and protonation on the activation energy of the thermal E/Z isomerization, it is never clearly rationalized in the text why the THI activation energy should not be affected by the polarity of the medium or by the presence/absence of the push-pull character. Despite their hypothesis being intuitively and qualitatively correct and their message can be grasped by the reader, I believe that the reader could benefit from a clear (possibly quantitative) statement that rationalize this point. Also considering the high level of the experimental and calculated data.

In a previous publication by our group, we describe an in-depth study on the influence of different solvent properties on the thermal helix inversion step of a structurally closely related motor (*Phys. Chem. Chem. Phys.* **2016**, *18*, 26725–26735.). No significant effect of solvent polarity was found. We have added this reference to the manuscript (ref. 67).

Furthermore, our calculated structures of **1_s** in gas and **(1-H)_s⁺** in MeCN (1.366 vs 1.360 Å) show no significant difference for the bond length of the central alkene suggesting that the steric clash between the two motor halves, which governs the activation barrier for thermal helix inversion, should be comparable. A sentence has been added to the discussion in the manuscript highlighting this point.

It would be nice to report, along with the enthalpy and entropy of activation, the Eyring plots for the THI process examined by NOESY (ESI section 4.5).

The Eyring plots have been added to Figure S19.

In table S4, the temperature of determination is needed only for the ΔG of activation. Enthalpy and entropy are extracted from Eyring analysis under the assumption of being temperature-independent. Therefore the notation ΔH(20°C) has no meaning in this context.

This mistake was corrected in Table S5 and Table S4 (note that the numbering of tables has changed).

In the caption of the NMR spectra should be reported the field of the instrument as well as the t_{mix} used for acquiring the NOESY spectra. These are particularly important when dealing with dynamic NMR.

This information has been added to the SI. Added text is highlighted in yellow.

In figure S10 and S11 signal “a” shifts upfield when irradiating at 530 nm, while it shifts downfield when irradiating at 470 nm. As the product of E/Z photoconversion is expected to be the same, what can this difference be ascribed to? Is this simply due to the solvent in which the spectra are recorded?

This phenomenon is, indeed, not caused by the change of irradiation wavelength but by the choice of solvent. The spectra in Figure S10 were recorded in toluene-*d*₈ whereas those in Figure S11 were recorded in DCM-*d*₂. In toluene-*d*₈ H_a is more shielded in the metastable isomer compared to the stable one, whereas the opposite effect is observed in DCM-*d*₂. We attribute this to different interactions of the solvents with the metastable isomer, potentially leading to different geometries of our compound.

Consistency should be used in the temperature units between the main text and the SI and within the SI, they are reported sometimes in Kelvin and sometimes Celsius.

This point has been addressed. Changes are highlighted in yellow.

At line 242 I would add the solvent explicitly, so it reads like "... 1m undergoes THI and not... after photoisomerization in hexane". At line 286 I believe the authors meant the "free energy of activation", which is a kinetic parameter that can be extracted by EXSY, and not the "standard free energy of formation", which is a thermodynamic parameter and to my understanding has no meaning in this context.

These points have been changed accordingly. Changes are highlighted in yellow.

REVIEWER COMMENTS

Reviewer #1 (Remarks to the Author):

The authors have fully addressed all my points. I therefore support publication of this beautiful work in the current form.

Reviewer #2 (Remarks to the Author):

In their response to my review of the original manuscript, the authors have clarified the minor issues that I had with this work. Accordingly, I am now happy to recommend this work for publication in Nat Commun.

Reviewer #3 (Remarks to the Author):

This reviewer is satisfied with the comments of the authors and the revised version of the manuscript. I thank the authors for addressing all the points.